# ENHANCING IMAGE RESTORATION TRANSFORMER WITH ADAPTIVE TOKEN DICTIONARY

## ABSTRACT

Image restoration is a classic computer vision problem that involves estimating high-quality (HQ) images from low-quality (LQ) ones. To compensate the information loss in the degradation process, prior knowledge of HQ image is indispensable. While deep neural networks (DNNs), especially Transformers for image restoration, have seen significant advancements in recent years, challenges still remain, particularly in the explicit incorporation of external priors, managing computational complexity, and tailoring generalized external priors to image specifics. To address these issues, we propose to enhance Transformer with **A**daptive **T**oken **D**ictionary (ATD), leading to a novel architecture which introduces a token dictionary to explicitly model external prior in the attention mechanism. The proposed ATD calculates the attention between the input features and the token dictionary, which integrates similar features on a global scale. Furthermore, we propose an adaptive dictionary refinement mechanism (ADR) to progressively customize the shared tokens to image specifics from shallow to deep layers. Crucially, benefiting from the condensed token dictionary, the computational complexity of the new attention mechanism is reduced from quadratic to linear with respect to the number of image tokens. This efficiency makes our network notably advantageous in constrained settings. Experimental results show that our method achieves best performance on various image restoration benchmark.

## 1 INTRODUCTION

The task of image restoration (IR) aims to recover clean high-quality (HQ) images from a solitary degraded low-quality (LQ) image or a sequence of such images, including image super-resolution, image denoising and JPEG compression artifact reduction. Since each LQ image may correspond to an infinite number of possible HQ images, image restoration is a classic ill-posed and challenging problem in the fields of computer vision and image processing. This practice is significant as it transcends the resolution and accuracy limitations of cost-effective sensors and enhances images produced by outdated equipment. The inherent information loss during the degradation process necessitates the incorporation of prior knowledge, specifically *external* and *internal* image priors, to supplement information for HQ image estimation. External priors refer to the generalized knowledge extracted from training datasets, whereas internal priors denote the image-specific information derived from the input image itself (Wang et al., 2015). The utilization and interplay of these external and internal image priors present numerous intriguing challenges and opportunities for exploration within the field.

In earlier research, various models such as Markov random field (He & Siu, 2011) and Dictionary Learning methods (Yang et al., 2010), were exploited to extract external priors explicitly from training datasets in a generative manner. The swift advancement of deep learning technologies in recent years has catalyzed an exponential increase in the application of deep neural network (DNN) models for image restoration. Pioneered by SRCNN (Dong et al., 2015) and DnCNN (Zhang et al., 2017a), convolutional neural network (CNN) based image restoration methods have emerged (Kim et al., 2016; Lim et al., 2017; Zhang et al., 2018a; Dai et al., 2020; Zhang et al., 2021). These methods directly learn the LQ-to-HQ mapping function with CNNs in a discriminative manner and implicitly embeds external prior knowledge of HQ image in the learned mapping functions. Recently, another family of DNNs, *i.e.*, Transformer-based image restoration networks have demonstrated their superiority to CNN-based networks, primarily by utilizing self-attention between image tokens to model internal

image priors (Liang et al., 2021; Chen et al., 2023; Li et al., 2023). Yet, how to explicitly embed external priors in DNN-based solutions for image restoration remains to be answered.

The second highly related issue about modelling internal image priors with Transformer-based networks is the computational complexity. The size of receptive field in vision transformers plays a critical role in capturing internal prior across an extensive range of patches (Liu et al., 2021; 2022). Despite achievements in image partitioning for self-attention, it continues to grapple with escalating computational complexity, growing quadratically with the number of input tokens and quadruply with the window size, which becomes particularly evident with increased sizes (Chen et al., 2023).

Moreover, although external priors offer a broad understanding of various image characteristics and patterns, each image possesses unique features and nuances that cannot be fully captured by these generalized external priors. By tailoring these priors to the specifics of the individual image, the network can better understand and leverage the unique properties of that image, enhancing the quality and accuracy of the resultant HR image. This adaptive approach allows for a more personalized and precise image reconstruction.

In summary, we try to address the following three research questions in this paper:

1. What methods can be employed to explicitly integrate external image priors into Transformer-based networks?
2. How can we reduce the computational complexity of Transformer-based networks while effectively modeling internal image priors for image restoration?
3. How can we tailor the incorporated external image prior to specific image characteristics?

Our solution is inspired by the classical dictionary learning methods, which models image prior with learned dictionary items. We propose to learn and adapt a token dictionary which seamlessly solves the three research questions. First of all, the external prior of the entire training dataset is condensed into the token dictionary during the training process. To exploit the external prior of token dictionary, we then implement an attention operation to select relevant dictionary keys and reconstruct enhanced features with their corresponding value tokens. Secondly, the complexity of the above token dictionary attention mechanism (Vaswani et al., 2017) is notably reduced to linear in relation to the number of image tokens. This allows us to efficiently apply the attention mechanism to enhance all the tokens in the image. Thirdly, we propose a refining strategy to adaptively fit the public dictionary to specific testing image. Specifically, the the attention map between token dictionary and image tokens encapsulates the similarity relationship between image tokens and the dictionary. By recurrently weighting the enhanced token with previous attention map, we can refine the public token dictionary based on the enhanced local structures of the input image. When combined with the original Swin Multi-head Self-Attention (SW-MSA) block (Liu et al., 2021; Liang et al., 2021), our proposed Adaptive Token Dictionary Cross-Attention (ATDCA) block enables the proposed network to balance use of internal and external image prior effectively.

Our contributions can be summarized as follows:

• We introduce a token dictionary learning method that incorporates external prior from the training dataset to augment the internal prior of existing self-attention-based IR approaches.
• We put forward a cross-block refining strategy that adaptively tailors the learned public token dictionary to a specific input image during the testing phase, enhancing IR results.
• By combining the proposed adaptive token dictionary attention with Swin self-attention, we achieve a balanced use of internal and external priors. This proposed method offers an improved balance between accuracy and computational load compared with the current state-of-the-art.

## 2 RELATED WORKS

**DNN-based Image Restoration.** SRCNN (Dong et al., 2015) was the first to use deep learning for single image super-resolution with a simple 3-layer neural network. DnCNN (Zhang et al., 2017a) is a pioneering work in image denoising. Since its development, many other works (Kim et al., 2016; Lim et al., 2017; Zhang et al., 2018a; 2019; Dai et al., 2020; Niu et al., 2020; Mei et al., 2021; Zhang et al., 2021) have explored a vast range of structures to boost performance. Among them, EDSR (Lim et al., 2017) and RDN (Zhang et al., 2018b) introduced new residual blocks with detailed connection designs, enhancing the capabilities of convolutional neural networks further.

With the quick growth of Transformer (Vaswani et al., 2017) in NLP, several works have enhanced performance using attention mechanisms. Since ViT (Dosovitskiy et al., 2020) and its variants (Liu et al., 2021; Chu et al., 2021; Wang et al., 2022) have shown the effectiveness of pure Transformer-based models in image classification, more works are exploring the potential of Transformer-based networks (Liang et al., 2021; Zamir et al., 2022; Zhang et al., 2023; Chen et al., 2023; Li et al., 2023) in image restoration, showing their superiority to CNN-based methods. These studies investigated a range of techniques to enhance the performance of image restoration transformers. The explored methods include window self-attention (Liang et al., 2021), channel self-attention (Zamir et al., 2022), and anchored self-attention (Li et al., 2023), among others. Additionally, pretraining on extensive datasets (Li et al., 2021), employing sparse attention (Zhang et al., 2023), and utilizing large window sizes (Chen et al., 2023) were strategies used to further boost performance.

**External Prior and Internal Prior Modeling.** Image restoration is an ill-posed problem due to information loss during the degradation process. Additional knowledge is needed to compensate for the information loss. In earlier studies, some methods were proposed to estimate the HQ image using the Bayesian framework. But in the past two decades, most image restoration methods learn the LQ to HQ mapping directly, embedding the prior of HQ image in the learned mapping functions.

Traditionally, two types of prior have been used for image restoration (Wang et al., 2015). One approach learns an **external prior** from a universal set of training data to predict the missing information for the HR image. Various functions, including local regression models (Timofte et al., 2015), coupled dictionary learning models (Yang et al., 2010), and deep neural networks (Dong et al., 2015), have been used to capture the external prior implicitly. Another approach searches for example patches from the input image itself to use the image's internal prior of cross-scale non-local self-similarity (Freedman & Fattal, 2011; Glasner et al., 2009; Shocher et al., 2018), providing relevant but limited references.

Recently, besides using internal prior for mapping function learning, researchers also proposed using **internal prior** by designing specific network architectures. The non-local layer (Wang et al., 2018) in CNNs benefits from the non-local prior of natural images. The self-attention block (Dosovitskiy et al., 2020) uses similarity between input tokens to combine input token features effectively. Both the non-local block (Zhang et al., 2019) and the self-attention block (Liang et al., 2021) enhance CNNs or MLPs for image restoration by balancing the use of external and internal priors.

This paper reveals that the self-attention block in Vision Transformer (Dosovitskiy et al., 2020) can also effectively model external prior. Inspired by conventional dictionary learning approaches, we introduce token dictionaries to model image external prior explicitly in Transformer-based image restoration network. We propose an adaptive strategy that updates the token dictionary based on the specific content of the input image, balancing the use of internal and external prior and delivering state-of-the-art image restoration results.

## 3 METHODOLOGY

### 3.1 MOTIVATION

In this subsection, we discuss how the dictionary learning based and self-attention based image restoration methods utilize external and internal prior to provide supplementary information for image restoration. Then we discuss how those two method motivates us to introduce external prior to Transformer-based methods with learned token dictionary.

**Dictionary Learning for Image Restoration.** Before the era of deep learning, dictionary learning methods play an important role in providing prior information for image restoration. Due to the limited computational resource, conventional dictionary learning based methods divide image into patches for modeling image local prior. Take image super-resolution for example. Denote $x \in \mathbb{R}^d$ as a vectorized image patch in the low-resolution(LR) image. To estimate the corresponding high-resolution(HR) patch $y \in \mathbb{R}^d$, Yang et al. (2010) decompose the signal by solving the sparse representation problem:

$$\alpha^* = argmin_{\alpha} \|x - D_L \alpha\|_2^2 + \lambda \|\alpha\|_1 \tag{1}$$

and reconstruct the HR patch with $D_H \alpha^*$; where $D_L \in \mathbb{R}^{d \times N}$ and $D_H \in \mathbb{R}^{d \times N}$ are the learned LR and HR dictionaries, and $N$ is the number of atoms in the dictionary. The coupled dictionaries $D_L$

and $\boldsymbol{D}_H$, summarize prior information of the external training dataset to compensate losing details in HR image.

**Vision Transformer for Image Restoration.** CNN-based image restoration methods learn spatially invariant convolution kernels from the training dataset to capture the LQ-to-HQ mapping. From a prior modeling perspective, local features of the input LQ image are enhanced based on external prior which were embedded in the convolution kernels in the training phase. On the other hand, the Transformer-based methods pay more attention to image internal priors and exploit similarity between tokens as weight to mutually enhance image features:

$$\text{Attention}(\boldsymbol{Q}, \boldsymbol{K}, \boldsymbol{V}) = \text{SoftMax}\left(\boldsymbol{Q}\boldsymbol{K}^T / \sqrt{d}\right)\boldsymbol{V}; \tag{2}$$

$\boldsymbol{Q} \in \mathbb{R}^{N \times d}$, $\boldsymbol{K} \in \mathbb{R}^{N \times d}$ and $\boldsymbol{V} \in \mathbb{R}^{N \times d}$ are linearly transformed from the input feature $\boldsymbol{X} \in \mathbb{R}^{N \times d}$ itself, $N$ is the token number and $d$ is the feature dimension. Due to the self-attentive processing philosophy, large window size plays a critical role in modeling internal prior of more patches. However, the complexity of self-attention computation increases quadratically with the number of input tokens, different strategies including shift-window (Liu et al., 2021; 2022; Liang et al., 2021; Conde et al., 2023), anchor attention (Li et al., 2023), and shifted crossed attention (Li et al., 2021) have been proposed to alleviate the limited window size issue of vision Transformer.

**Token Dictionary: Empower Attention Block with External Prior.** After reviewing the above contents, we found that the decomposition and reconstruction idea of dictionary learning based image restoration is similar to the process of self-attention computation. Specifically, the above method in Eq. 1 solves the sparse representation model to find similar LR dictionary atoms and reconstruct HR signal with the corresponding HR dictionary atoms; while the attention-based methods use normalized point product operation to determine attention values for combining value tokens. The above observation implies that the idea of using coupled dictionary $\boldsymbol{D}_L$ and $\boldsymbol{D}_H$ to introduce external prior can be easily incorporated into the Transformer framework. By learning an extra token dictionary instead of generating key and value tokens from input image, we can summarize external prior of training dataset for better restoration.

In the following sub-sections, we will firstly introduce how we design our Token Dictionary Cross-Attention (TDCA) block to introduce external prior into the Transformer framework. Then, we further improve our proposed TDCA by proposing a refine strategy to adaptively fit the public dictionary to each specific input image.

## 3.2 TOKEN DICTIONARY CROSS-ATTENTION

In this subsection, we introduce details of our proposed token dictionary cross-attention block.

In comparison to the existing MSA which generates query, key and value tokens by the input feature itself. We aim to introduce an extra dictionary $\boldsymbol{D} \in \mathbb{R}^{M \times d}$ to summarize external prior from the training data. We use the learned token dictionary $\boldsymbol{D}$ to generate the Key Dictionary $\boldsymbol{K}_D$ and the Value Dictionary $\boldsymbol{V}_D$ and use the input feature $\boldsymbol{X} \in \mathbb{R}^{N \times d}$ to generate query tokens:

$$\boldsymbol{Q}_X = \boldsymbol{X}\boldsymbol{W}^Q, \qquad \boldsymbol{K}_D = \boldsymbol{D}\boldsymbol{W}^K, \qquad \boldsymbol{V}_D = \boldsymbol{D}\boldsymbol{W}^V, \tag{3}$$

where $M \ll N$, and $W^Q \in \mathbb{R}^{d \times d/r}$, $W^K \in \mathbb{R}^{d \times d/r}$ and $W^V \in \mathbb{R}^{d \times d}$ are linear transforms for Query Tokens, Key Dictionary Tokens and Value Dictionary Tokens, respectively. Note that the feature dimensions of query tokens and key dictionary tokens are reduced to $1/r$ for decreasing parameters and save computational resource consumption, where $r$ is the reduction ratio. Then, we use the key dictionary and value dictionary to enhance query tokens via cross-attention calculation:

$$\text{TDCA}(\boldsymbol{Q}_X, \boldsymbol{K}_D, \boldsymbol{V}_D) = \text{SoftMax}\left(\boldsymbol{S}/\tau\right)\boldsymbol{V}_D, \quad \text{where } \boldsymbol{S} = \text{Sim}_{\cos}(\boldsymbol{Q}_X, \boldsymbol{K}_D). \tag{4}$$

In Eq. 4, $\tau$ is a learnable parameter for adjusting the range of similarity value. $\text{Sim}_{\cos}(\cdot, \cdot)$ represents calculating Cosine Similarity between two tokens, and $\boldsymbol{S} \in \mathbb{R}^{N \times M}$ is the similarity map between Query image tokens and Key dictionary tokens. We use the normalized Cosine distance instead of dot product operation in MSA because we want each token in the Dictionary to have equal opportunity to be selected, the similar magnitude normalization operation is commonly used in previous dictionary learning works.

The above TDCA operation firstly selects similar tokens in Key Dictionary tokens, which is similar to the sparse representation process in Eq. 1 to obtain representation coefficients; then, TDCA

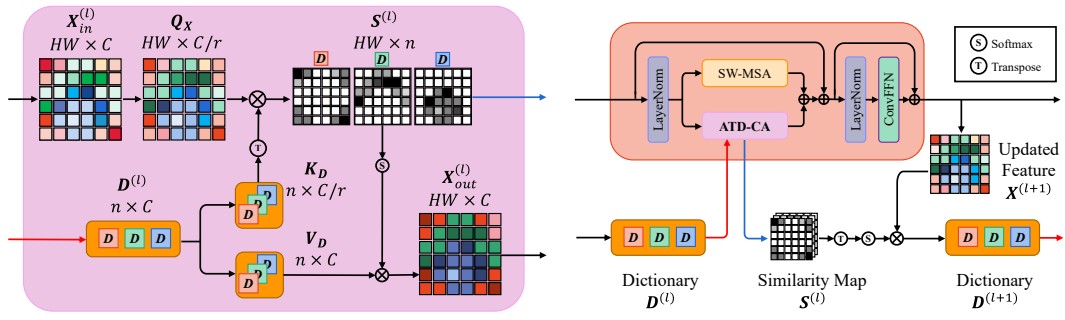

(a) Token Dictionary Cross-Attention  (b) Transformer Layer and Adaptive Dictionary Refinement

Figure 1: The proposed Token Dictionary Cross-Attention Block (TDCA) block and the Adaptive Dictionary Refinement (ADR) strategy. More details of our proposed TDCA and ADR can be found in Sec. 3.2 and 3.3, respectively.

utilizes the similarity values to combine the corresponding Value Dictionary tokens, which is the same as reconstructing HQ patch with HQ dictionary atoms and representation coefficients. By this way, our TDCA is able to embed the external prior into learned dictionary for enhancing the input image feature. We will validate the effectiveness of using token dictionary to provide external prior information in our ablation study at Sec. 4.1.

### 3.3 ADAPTIVE TOKEN DICTIONARY REFINEMENT

In the previous subsection, we have presented how we could introduce an extra token dictionary to supply external prior for image restoration Transformer. Since the image feature in each block will be projected to different feature space by Multi-Layer Perceptrons (MLPs), we need to learn Token Dictionary for each block to provide external prior in each specific feature space. This will lead to a large number of extra parameters. In this subsection, we introduce an adaptive refining strategy which refines token dictionary of the previous block based on the similarity map and the updated features.

To introduce the proposed adaptive refining strategy, we set up the block index $(l)$ for the input features and token dictionary, i.e. $\boldsymbol{X}^{(l)}$ and $\boldsymbol{D}^{(l)}$ denotes the input feature and token dictionary of the $l$-th block, respectively. We learn token dictionary for the first block $\boldsymbol{D}^{(0)}$ to introduce external prior as introduced in Sec. 3.2. While, for each token in the dictionary $\left\{\boldsymbol{d}_i^{(l)}\right\}_{i=1,\dots,M}$ of the $l$-th block, we select the corresponding similar tokens in the image to refine it. To be more specific, we denote $\boldsymbol{s}_i^{(l-1)}$ as the $i$-th column of similarity map $\boldsymbol{S}^{(l-1)}$, it contains the distance between $\boldsymbol{d}_i^{(l-1)}$ and all the $N$ query tokens $\boldsymbol{X}^{(l-1)}$. Therefore, based on $\boldsymbol{s}_i^{(l-1)}$, we are able to select the corresponding enhanced tokens $\boldsymbol{X}^{(l)}$ to reconstruct the new token dictionary $\boldsymbol{d}_i^{(l)}$:

$$\boldsymbol{D}^{(l)} = \text{SoftMax}\left(\boldsymbol{S}^{(l-1)^T}/\sigma\right)\boldsymbol{X}^{(l)}, \tag{5}$$

where $\sigma$ is a learnable scaling value to adjust the range of similarity map.

Thanks to the linear complexity of the proposed TDCA with the number of image tokens, we do not need to divide the image into windows and $\boldsymbol{X}^{(l)}$ represents all the image tokens. A visualization of the intermediate similarity map can be found in Fig. 2, it can be easily found that different similarity vectors $\boldsymbol{s}_i$ captures different types of textures in the input image. Starting from the initial token dictionary $\boldsymbol{D}^{(0)}$, which introduces external prior into the network, our adaptive refining strategy gradually select relevant tokens from the whole image to refine the dictionary. The refined dictionary could cross the boundary of self-attention window to summarize the typical local structures of the whole image, and consequently, improve image feature with global information.

### 3.4 THE OVERALL NETWORK ARCHITECTURE

Having our proposed Token Dictionary Cross-Attention (TDCA) block and the Adaptive Dictionary Refinement (ADR) strategy in Sec. 3.2 and 3.3, we are able to establish our Adaptive Token Dictionary



|     |     |     |     |
| :-: | :-: | :-: | :-: |
| (a) | (b) | (c) | (d) |

Figure 2: Visualization of similarity vectors $\boldsymbol{s}_i^{(l)}$ of an image in the Urban 100 dataset. (b), (c) and (d) show similarity map between all the input tokens and three tokens in our token dictionary. The similarity map clearly shows that different tokens in the dictionary could detect different types of local structures. Therefore, we are able to adaptively fit the public dictionary to specific testing image by summarizing image structures based on the similarity map.

(ATD) network for image restoration. Given an input LQ image, we firstly utilize a $3\times3$ convolution layer to extract shallow features. Then, the shallow features are fed into several ATD blocks in specific architecture depending on the task. Each ATD block contains $N_{Trans}$ transformer layers. The transformer layer contains our proposed Adaptive Token Dictionary Cross-Attention (ATDCA) and a Shift Window-based MSA(SW-MSA) (Liang et al., 2021; Liu et al., 2021), the two kinds of attention blocks process the input feature in parallel and the final features are combined by a summation operation. In addition to the attention block, our transformer layer also utilize LayerNorm and ConvFFN layers, which have been commonly utilized in other Transformer-based architectures. After the ATD blocks, we utilize an extra convolution layer (followed with pixel shuffle operation for SR task) to generate the final HQ estimation. For image SR, ATD blocks are connected in sequence and we provide an illustration of our network architecture in Fig. 3. For image denoising and JPEG compression artifact reduction, an encoder-decoder architecture is employed following previous works as shown in Fig. 6 in Appendix.

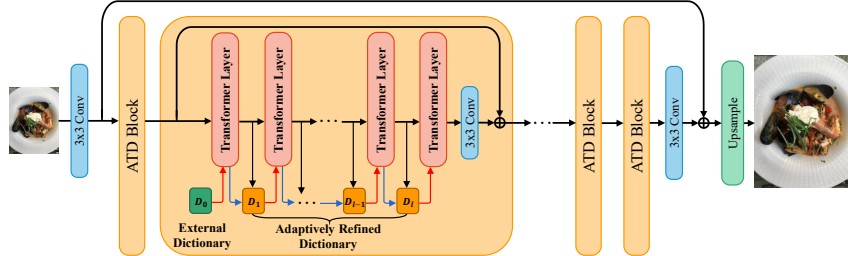

Figure 3: The overall architecture of the proposed ATD network for image super-resolution. The illustration of the ATD-U (encoder-decoder architecture for image denoising and JPEG compression artifact reduction) is presented in the Fig. 6.

## 4 EXPERIMENTS

In this section we provide experimental results on several image restoration tasks including 1) image super-resolution, 2) image denoising, and 3) JPEG compression artifacts reduction. Three models with different architecture and parameter size are built for different tasks. We establish ATD as well as its tiny version ATD-light to tackle super-resolution problem, while ATD-U is designed for denoising and JPEG compression artifact reduction task. Due to page limitations, the detailed experimental settings are presented in the Appendix.

### 4.1 ABLATION STUDY

Before comparing our model with state-of-the-art methods, we firstly conduct ablation studies to validate our design choices. We conduct ablation studies on re-scaled ATD-light model and train all the models for 500k iterations on DIV2K (Timofte et al., 2017) dataset. We then evaluate them on the Urban100 (Huang et al., 2015) benchmark.

**Effects of External Token Dictionary and Adaptive Token Dictionary.** In order to show the effectiveness of the proposed adaptive token dictionary cross-attention (AT-DCA) approach, we establish three models and compare their capability for image SR. The first model is our final model **ATD**, which learns external token dictionary from training data and utilizes refinement strategy to update token dictionary in each ATD blocks. To demonstrate the effectiveness of learned token dictionary, we present a baseline model **ATD(-t)** which do not learn any token

Table 1: Ablation study on the effects of each component. More details of the experimental settings can be found in our Ablation study section.

| model | TD | ADR | PSNR(dB) | SSIM |
|---|---|---|---|---|
| ATD(-t) | | | 26.67 | 0.8038 |
| ATD(-a) | √ | | 26.72 | 0.8044 |
| ATD | √ | √ | 26.77 | 0.8051 |

dictionary and only adopt SW-MSA block to process image features. Then, to analyze the effect of our adaptive dictionary refinement strategy, we establish another model, i.e. the **ATD(-a)**, which directly learns external token dictionary for each Transformer layer without applying adaptive refinement operation. The ablation results can be found in Table 1. The results clearly show that the learned token dictionary could provide external information for better SR, and the refinement strategy is able to further enhance the learned public dictionary while reducing the size of dictionary.

Table 2: Quantitative comparison (PSNR/SSIM) with state-of-the-art methods on **lightweight SR** and **classical SR** task. Best and second best results are colored with red and blue. More experimental details can be found in the main text.

| Method | Params | Scale | Set5 PSNR | Set5 SSIM | Set14 PSNR | Set14 SSIM | BSD100 PSNR | BSD100 SSIM | Urban100 PSNR | Urban100 SSIM | Manga109 PSNR | Manga109 SSIM |
|---|---|---|---|---|---|---|---|---|---|---|---|---|
| CARN (Ahn et al., 2018) | ×2 | 1,592K | 37.76 | 0.9590 | 33.52 | 0.9166 | 32.09 | 0.8978 | 31.92 | 0.9256 | 38.36 | 0.9765 |
| IMDN (Hui et al., 2019) | ×2 | 694K | 38.00 | 0.9605 | 33.63 | 0.9177 | 32.19 | 0.8996 | 32.17 | 0.9283 | 38.88 | 0.9774 |
| LAPAR-A (Li et al., 2020) | ×2 | 548K | 38.01 | 0.9605 | 33.62 | 0.9183 | 32.19 | 0.8999 | 32.10 | 0.9283 | 38.67 | 0.9772 |
| LatticeNet (Luo et al., 2020) | ×2 | 756K | 38.15 | 0.9610 | 33.78 | 0.9193 | 32.25 | 0.9005 | 32.43 | 0.9302 | - | - |
| SwinIR-light (Liang et al., 2021) | ×2 | 910K | 38.14 | 0.9611 | 33.86 | 0.9206 | 32.31 | 0.9012 | 32.76 | 0.9340 | 39.12 | 0.9783 |
| ELAN (Zhang et al., 2022) | ×2 | 582K | 38.17 | 0.9611 | 33.94 | 0.9207 | 32.30 | 0.9012 | 32.76 | 0.9340 | 39.11 | 0.9782 |
| SwinIR-NG (Choi et al., 2022) | ×2 | 1181K | 38.17 | 0.9612 | 33.94 | 0.9205 | 32.31 | 0.9013 | 32.78 | 0.9340 | 39.20 | 0.9781 |
| OmniSR (Wang et al., 2023) | ×2 | 772K | 38.22 | 0.9613 | 33.98 | 0.9210 | 32.36 | 0.9020 | 33.05 | 0.9363 | 39.28 | 0.9784 |
| **ATD-light** (ours) | ×2 | 757K | 38.27 | 0.9615 | 34.05 | 0.9218 | 32.38 | 0.9022 | 33.22 | 0.9380 | 39.35 | 0.9781 |
| EDSR (Lim et al., 2017) | ×2 | 42.6M | 38.11 | 0.9602 | 33.92 | 0.9195 | 32.32 | 0.9013 | 32.93 | 0.9351 | 39.10 | 0.9773 |
| RCAN (Zhang et al., 2018a) | ×2 | 15.4M | 38.27 | 0.9614 | 34.12 | 0.9216 | 32.41 | 0.9027 | 33.34 | 0.9384 | 39.44 | 0.9786 |
| SAN (Dai et al., 2020) | ×2 | 15.7M | 38.31 | 0.9620 | 34.07 | 0.9213 | 32.42 | 0.9028 | 33.10 | 0.9370 | 39.32 | 0.9792 |
| HAN (Niu et al., 2020) | ×2 | 63.6M | 38.27 | 0.9614 | 34.16 | 0.9217 | 32.41 | 0.9027 | 33.35 | 0.9385 | 39.46 | 0.9785 |
| IPT (Chen et al., 2020) | ×2 | 115M | 38.37 | - | 34.43 | - | 32.48 | - | 33.76 | - | - | - |
| SwinIR (Liang et al., 2021) | ×2 | 11.8M | 38.42 | 0.9623 | 34.46 | 0.9250 | 32.53 | 0.9041 | 33.81 | 0.9433 | 39.92 | 0.9797 |
| EDT (Li et al., 2021) | ×2 | 11.5M | 38.45 | 0.9624 | 34.57 | 0.9258 | 32.52 | 0.9041 | 33.80 | 0.9425 | 39.93 | 0.9800 |
| CAT-A (Chen et al., 2022) | ×2 | 16.5M | 38.51 | 0.9626 | 34.78 | 0.9265 | 32.59 | 0.9047 | 34.26 | 0.9440 | 40.10 | 0.9805 |
| ART (Zhang et al., 2023) | ×2 | 16.4M | 38.56 | 0.9629 | 34.59 | 0.9267 | 32.58 | 0.9048 | 34.30 | 0.9452 | 40.24 | 0.9808 |
| HAT (Chen et al., 2023) | ×2 | 20.6M | 38.63 | 0.9630 | 34.86 | 0.9274 | 32.62 | 0.9053 | 34.45 | 0.9466 | 40.26 | 0.9809 |
| **ATD** (ours) | ×2 | 18.7M | 38.61 | 0.9630 | 34.77 | 0.9271 | 32.63 | 0.9054 | 34.49 | 0.9470 | 40.33 | 0.9808 |
| CARN (Ahn et al., 2018) | ×4 | 1,592K | 32.13 | 0.8937 | 28.60 | 0.7806 | 27.58 | 0.7349 | 26.07 | 0.7837 | 30.47 | 0.9084 |
| IMDN (Hui et al., 2019) | ×4 | 715K | 32.21 | 0.8948 | 28.58 | 0.7811 | 27.56 | 0.7353 | 26.04 | 0.7838 | 30.45 | 0.9075 |
| LAPAR-A (Li et al., 2020) | ×4 | 659K | 32.15 | 0.8944 | 28.61 | 0.7818 | 27.61 | 0.7366 | 26.14 | 0.7871 | 30.42 | 0.9074 |
| LatticeNet (Luo et al., 2020) | ×4 | 777K | 32.30 | 0.8962 | 28.68 | 0.7830 | 27.62 | 0.7367 | 26.25 | 0.7873 | - | - |
| SwinIR-light (Liang et al., 2021) | ×4 | 930K | 32.44 | 0.8976 | 28.77 | 0.7858 | 27.69 | 0.7406 | 26.47 | 0.7980 | 30.92 | 0.9151 |
| ELAN (Zhang et al., 2022) | ×4 | 582K | 32.43 | 0.8975 | 28.78 | 0.7858 | 27.69 | 0.7406 | 26.54 | 0.7982 | 30.92 | 0.9150 |
| SwinIR-NG (Choi et al., 2022) | ×4 | 1201K | 32.44 | 0.8980 | 28.83 | 0.7870 | 27.73 | 0.7418 | 26.61 | 0.8010 | 31.09 | 0.9161 |
| OmniSR (Wang et al., 2023) | ×4 | 792K | 32.49 | 0.8988 | 28.78 | 0.7859 | 27.71 | 0.7415 | 26.65 | 0.8018 | 31.02 | 0.9151 |
| **ATD-light** (ours) | ×4 | 772K | 32.50 | 0.8988 | 28.86 | 0.7884 | 27.76 | 0.7431 | 26.89 | 0.8097 | 31.29 | 0.9184 |
| EDSR (Lim et al., 2017) | ×4 | 43.0M | 32.46 | 0.8968 | 28.80 | 0.7876 | 27.71 | 0.7420 | 26.64 | 0.8033 | 31.02 | 0.9148 |
| RCAN (Zhang et al., 2018a) | ×4 | 15.6M | 32.63 | 0.9002 | 28.87 | 0.7889 | 27.77 | 0.7436 | 26.82 | 0.8087 | 31.22 | 0.9173 |
| SAN (Dai et al., 2020) | ×4 | 15.9M | 32.64 | 0.9003 | 28.92 | 0.7888 | 27.78 | 0.7436 | 26.79 | 0.8068 | 31.18 | 0.9169 |
| HAN (Niu et al., 2020) | ×4 | 64.2M | 32.64 | 0.9002 | 28.90 | 0.7890 | 27.80 | 0.7442 | 26.85 | 0.8094 | 31.42 | 0.9177 |
| IPT (Chen et al., 2020) | ×4 | 116M | 32.64 | - | 29.01 | - | 27.82 | - | 27.26 | - | - | - |
| SwinIR (Liang et al., 2021) | ×4 | 11.9M | 32.92 | 0.9044 | 29.09 | 0.7950 | 27.92 | 0.7489 | 27.45 | 0.8254 | 32.03 | 0.9260 |
| EDT (Li et al., 2021) | ×4 | 11.6M | 32.82 | 0.9031 | 29.09 | 0.7939 | 27.91 | 0.7483 | 27.46 | 0.8246 | 32.05 | 0.9254 |
| CAT-A (Chen et al., 2022) | ×4 | 16.6M | 33.08 | 0.9052 | 29.18 | 0.7960 | 27.99 | 0.7510 | 27.89 | 0.8339 | 32.39 | 0.9285 |
| ART (Zhang et al., 2023) | ×4 | 16.6M | 33.04 | 0.9051 | 29.16 | 0.7958 | 27.97 | 0.7510 | 27.97 | 0.8321 | 32.31 | 0.9283 |
| HAT (Chen et al., 2023) | ×4 | 20.8M | 33.04 | 0.9056 | 29.23 | 0.7973 | 28.00 | 0.7517 | 27.97 | 0.8368 | 32.48 | 0.9292 |
| **ATD** (ours) | ×4 | 18.9M | 33.07 | 0.9061 | 29.20 | 0.7979 | 28.01 | 0.7518 | 27.98 | 0.8374 | 32.49 | 0.9292 |

## 4.2 IMAGE SUPER-RESOLUTION

We firstly evaluate the proposed ATD method on the image super-resolution task. We choose Set5 (Bevilacqua et al., 2012), Set14 (Zeyde et al., 2012), BSD100 (Martin et al., 2002), Urban100 (Huang et al., 2015), and Manga109 (Matsui et al., 2016) as evaluation datasets and compare with recent state-of-the-art SR methods. For the task of lightweight SR, we compare our method with CARN (Ahn et al., 2018), IMDN (Hui et al., 2019), LAPAR (Li et al., 2020), LatticeNet (Luo et al., 2020), SwinIR (Liang et al., 2021), ELAN (Zhang et al., 2022) and OmniSR (Wang et al., 2023).

As can be found in Table 2, the proposed ATD-light achieved better results with OmniSR (Wang et al., 2023) in both ×2 and ×4 zooming factors across most of benchmark datasets. Our ATD-light outperform recently proposed light-weight method OmniSR by a large margin (0.27dB) on the anga109 dataset. e believe this is because more information was lost during the downscaling process for cases with large zoom factors. With the learned and refined token dictionary, our ATD-light model is able to make better use of external prior for recovering HR details in challenging conditions.

We further compare our method with state-of-the-art SR methods: EDSR (Lim et al., 2017), RCAN (Zhang et al., 2018a), SAN (Dai et al., 2020), HAN (Niu et al., 2020), CSNLN (Mei et al., 2020), IPT (Chen et al., 2020), SwinIR (Liang et al., 2021), CAT (Chen et al., 2022), ART (Zhang et al., 2023), HAT (Chen et al., 2023). With about 10% less number of parameters(18.7M vs. 20.8M), the proposed ATD model still outperforms HAT (Chen et al., 2023). We achieved +0.07dB PSNR gain on the ×2 Manga109 dataset.

Some visual examples by different methods can be found in Fig. 4. The images in Fig. 4 clearly demonstrate our advantages in recovering sharp edges and clean textures. More visual examples can be found in the Appendix A.3.

Table 3: Quantitative comparison (PSNR) with state-of-the-art methods on **grayscale image denoising**. Best and second best results are colored with red and blue. More experimental details can be found in the main text.

| Dataset | $\sigma$ | BM3D | DnCNN | IRCNN | RNAN | RDN | DRUNet | SwinIR | Restormer | **ATD-U** (ours) |
|---------|---|-------|-------|-------|-------|-------|--------|--------|-----------|-----------------|
| **Set12** | 15 | 32.37 | 32.86 | 32.76 | - | - | 33.25 | 33.36 | 33.42 | 33.47 |
|  | 25 | 29.97 | 30.44 | 30.37 | - | - | 30.94 | 31.01 | 31.08 | 31.16 |
|  | 50 | 26.72 | 27.18 | 27.12 | 27.70 | 27.60 | 27.90 | 27.91 | 28.00 | 28.09 |
| **BSD68** | 15 | 31.08 | 31.73 | 31.63 | - | - | 31.91 | 31.97 | 31.96 | 31.97 |
|  | 25 | 28.57 | 29.23 | 29.15 | - | - | 29.48 | 29.50 | 29.52 | 29.51 |
|  | 50 | 25.60 | 26.23 | 26.19 | 26.48 | 26.41 | 26.59 | 26.58 | 26.62 | 26.54 |
| **Urban100** | 15 | 32.35 | 32.64 | 32.46 | - | - | 33.44 | 33.70 | 33.79 | 34.05 |
|  | 25 | 29.70 | 29.95 | 29.80 | - | - | 31.11 | 31.30 | 31.46 | 31.83 |
|  | 50 | 25.95 | 26.26 | 26.22 | 27.65 | 27.40 | 27.96 | 27.98 | 28.29 | 28.81 |

### 4.3 IMAGE DENOISING

We build ATD-U based on encoder-decoder architecture following Zamir et al. (2022) for grayscale image denoising task and make comparison with recent state-of-the-art methods: DnCNN (Zhang et al., 2017a), IRCNN (Zhang et al., 2017b), RNAN (Zhang et al., 2019), RDN (Zhang et al., 2018b), DRUNet (Zhang et al., 2021), SwinIR (Liang et al., 2021), Restormer (Zamir et al., 2022) on Set12 (Zhang et al., 2017a), BSD68 (Martin et al., 2001) and Urban100 (Huang et al., 2015) datasets.

The quantitative results are shown in Table 3. Our ATD-U outperforms Restormer by a large margin up to 0.52dB on Urban100 benchmark under challenging noise level of 50 with 10% smaller model size (23.5M) compared to Restormer(26.1M). These comparisons illustrate the strong power of external prior when restoring image from severe noise. ATD-U also exhibits comparable and better performance on BSD68 and other datasets. These results indicate the powerful capacity of ATD-U to mitigate noise in grayscale images. We provide some visual examples in Fig. 4. These comparisons illustrate that ATD-U possess the ability to restore cleaner image from heavy noise pollution while resulting in less artifacts. More visual comparisons can be found in the Appendix A.3.

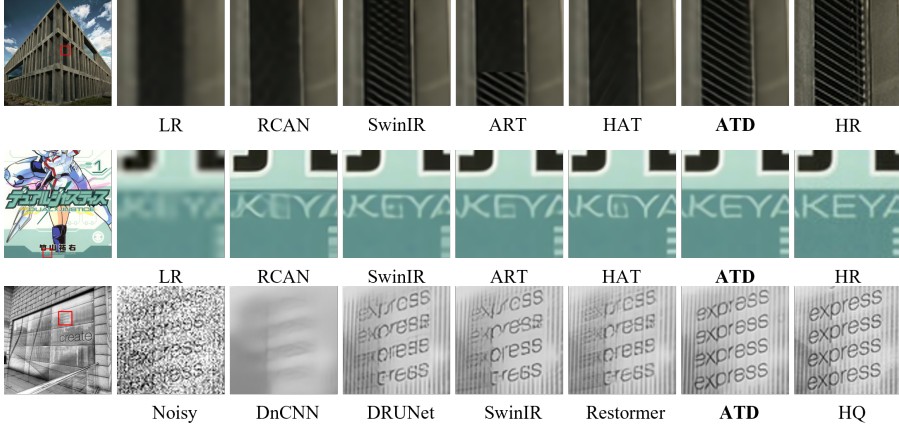

Figure 4: Visual comparisons of **image super-resolution** methods on image Urban100-"img027", Manga109-"DualJustice" and **grayscale image denoising** methods on image Urban100-"img060"

### 4.4 JPEG COMPRESSION ARTIFACT REDUCTION

Same as image denoising task, we employ ATD-U model on JPEG compression artifact reduction task and make comparison with recent state-of-the-art methods: DnCNN (Zhang et al., 2017a), RNAN (Zhang et al., 2019), RDN (Zhang et al., 2018b), DRUNet (Zhang et al., 2021), SwinIR (Liang et al., 2021), ART (Zhang et al., 2023). We choose Classic5 (Foi et al., 2007), LIVE1 (Sheikh et al., 2006) and Urban100 (Huang et al., 2015) as evaluation datasets. The experimental results are presented in Table 4. Our ATD-U achieves better performance compared to previous methods SwinIR and ART. ATD-U obtains up to 0.25dB gain on Urban100 under challenging compression quality factor(QF=10). These results manifest that ATD-U also has strong artifact removal ability for JPEG compression artifact reduction.

Table 4: Quantitative comparison (PSNR/SSIM) with state-of-the-art methods on **JPEG compression artifact reduction**. Best and second best results are colored with red and blue. More experimental details can be found in the main text.

| | | Grayscale | | | | | | | | | | | | Color | | | | | |
|---|---|---|---|---|---|---|---|---|---|---|---|---|---|---|---|---|---|---|---|
| Set | QF | DnCNN | | DRUNet | | SwinIR | | ART | | ATD-U (ours) | | Set | QF | DRUNet | | SwinIR | | ATD-U (ours) | |
| | | PSNR | SSIM | PSNR | SSIM | PSNR | SSIM | PSNR | SSIM | PSNR | SSIM | | | PSNR | SSIM | PSNR | SSIM | PSNR | SSIM |
| Classic5 | 10 | 29.40 | 0.8013 | 30.16 | 0.8234 | 30.27 | 0.8249 | 30.27 | 0.8258 | 30.38 | 0.8272 | Classic5 | 10 | 27.47 | 0.8045 | 28.06 | 0.8129 | 28.32 | 0.8179 |
| | 20 | 31.63 | 0.8596 | 32.39 | 0.8734 | 32.52 | 0.8748 | - | - | 32.60 | 0.8751 | | 20 | 30.29 | 0.8743 | 30.44 | 0.8768 | 30.58 | 0.8787 |
| | 30 | 32.91 | 0.8855 | 33.59 | 0.8949 | 33.73 | 0.8961 | 33.74 | 0.8964 | 33.80 | 0.8962 | | 30 | 31.64 | 0.9020 | 31.81 | 0.9040 | 31.97 | 0.9059 |
| | 40 | 33.77 | 0.8993 | 34.41 | 0.9075 | 34.52 | 0.9082 | 34.55 | 0.9086 | 34.59 | 0.9082 | | 40 | 32.56 | 0.9174 | 32.75 | 0.9193 | 32.89 | 0.9205 |
| LIVE1 | 10 | 29.19 | 0.8172 | 29.79 | 0.8278 | 29.86 | 0.8287 | 29.89 | 0.8300 | 29.94 | 0.8371 | BSD500 | 10 | 27.62 | 0.8001 | 28.22 | 0.8075 | 28.32 | 0.8083 |
| | 20 | 31.59 | 0.8848 | 32.17 | 0.8899 | 32.25 | 0.8909 | - | - | 32.31 | 0.8949 | | 20 | 30.39 | 0.8711 | 30.54 | 0.8739 | 30.54 | 0.8730 |
| | 30 | 32.98 | 0.9167 | 33.59 | 0.9166 | 33.69 | 0.9174 | 33.71 | 0.9178 | 33.72 | 0.9226 | | 30 | 31.73 | 0.9003 | 31.90 | 0.9025 | 31.90 | 0.9020 |
| | 40 | 33.96 | 0.9294 | 34.58 | 0.9312 | 34.67 | 0.9317 | 34.70 | 0.9322 | 34.70 | 0.9342 | | 40 | 32.66 | 0.9168 | 32.84 | 0.9189 | 32.80 | 0.9178 |
| Urban100 | 10 | 28.54 | 0.8484 | 30.31 | 0.8745 | 30.55 | 0.8835 | 30.87 | 0.8894 | 31.12 | 0.8935 | Urban100 | 10 | 27.10 | 0.8400 | 28.18 | 0.8586 | 29.07 | 0.8726 |
| | 20 | 31.01 | 0.9050 | 32.81 | 0.9241 | 33.12 | 0.9190 | - | - | 33.52 | 0.9271 | | 20 | 30.17 | 0.8991 | 30.53 | 0.9030 | 31.14 | 0.9094 |
| | 30 | 32.47 | 0.9312 | 34.23 | 0.9414 | 34.58 | 0.9417 | 34.81 | 0.9442 | 34.85 | 0.9458 | | 30 | 31.49 | 0.9189 | 31.87 | 0.9219 | 32.48 | 0.9271 |
| | 40 | 33.49 | 0.9412 | 35.20 | 0.9547 | 35.50 | 0.9515 | 35.73 | 0.9553 | 35.73 | 0.9573 | | 40 | 32.36 | 0.9301 | 32.75 | 0.9329 | 33.27 | 0.9365 |

### 4.5 COMPUTATIONAL BURDEN ANALYSIS

We further analyze the computational burden of the proposed ATD model. In Fig. 5, we present the image restoration accuracy (PSNR) and computational consumption of recent state-of-the-art models on the image SR and denoising tasks. The figures clearly demonstrate that the proposed ATD model helps the network to achieve a better trade-off between restoration accuracy and

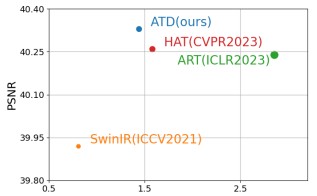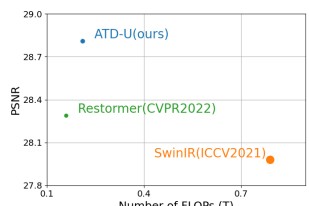

Figure 5: Comparison of PSNR and FLOPs by our model and S.O.T.A approaches (SR: Manga109 × 2; Denoise: Urban100 $\sigma = 50$), FLOPs are calculated on input size of $256 \times 256$.

efficiency. Our ATD method achieves better SR results with 10% and 50% less computations than the HAT and ART model, respectively. Moreover, with only 10% more computations, our ATD-U model is able to improve the denoising performance of Restormer by a large margin (0.52dB). In comparison with the baseline SwinIR approach, ATD-U surpasses SwinIR by 0.83dB with 3.7× less FLOPs.

## 5 CONCLUSION

In this paper, we proposed a new Transformer-based image restoration network. Inspired by the traditional dictionary learning methods, we proposed to learn token dictionaries to provide external supplementary information for estimating the missing high-quality details. We further proposed an adaptive dictionary refinement strategy which could use similarity map of the previous block to refine the externally learned dictionary, making it better fit the content of specific input image. We conducted ablation studies to demonstrate the effectiveness of the proposed token dictionary and adaptive refinement strategy. Extensive experimental results on plenty of benchmark datasets have been presented, our method achieved state-of-the-art results on image super-resolution, grayscale image denoising and JPEG compression artifact reduction.

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

# A APPENDIX

## A.1 EXPERIMENTAL SETTING DETAILS.

### A.1.1 NETWORK ARCHITECTURE SETTING.

**ATD** For image super-resolution task, we establish ATD model that employs a sequence of ATD blocks as its backbone. There are 6 ATD blocks in total, each comprising six transformer layers with a channel number of 200. We establish 512 external tokens for our dictionary in ATD model and use a reduction rate $r$ of 10. Each external token has 200 feature dimensions as in the SW-MSA branch, and each external token dictionary is randomly initialized as a tensor with shape of $[512, 200]$ in normal distribution.

**ATD-light** ATD-light is a tiny version of ATD which reduce feature dimensions to 48 for lightweight SR task. The number of dictionary token is decreased to 64 and we also adjust the reduction rate $r$ to 4 for keeping enough in similarity calculation.

**ATD-U** For image denoising and JPEG compression artifact reduction, we employ a 4-level encoder-decoder architecture on ATD-U following Restormer (Zamir et al., 2022). An illustration of ATD-U architecture is presented in Fig. 6. We set the number of Transformer layers for each level as [4, 6, 6, 8], while the number of channel and reduction rate $r$ are set as [48, 96, 192, 384] and [3, 6, 12, 24]. The parameter setting of refinement block is the same as level-1 encoder.

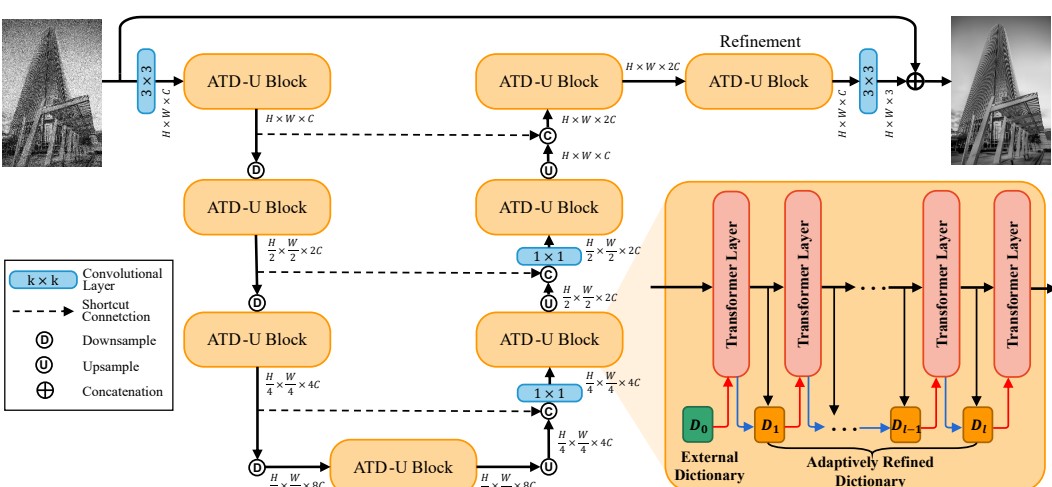

Figure 6: The overall architecture of the proposed ATD-U network for image denoising and JPEG compression artifact reduction.

### A.1.2 OTHER DETAILS ABOUT ARCHITECTURE.

**Hard threshold operation.** When applying adaptive token dictionary refinement as mentioned in Eq. 5, we combine the similarity map and features to generate refined token dictionary. Practically, instead of directly using the similarity map from the previous block, we adopt an additional hard thresholding operation:

$$\boldsymbol{S}_\delta(i,j) = \text{HardThresholding}\left(\boldsymbol{S}(i,j); \delta\right) = \begin{cases} \boldsymbol{S}(i,j) & \text{if } \boldsymbol{S}(i,j) \geq \delta \\ -\infty & \text{if } \boldsymbol{S}(i,j) < \delta \end{cases}, \tag{6}$$

where $\delta$ is the threshold in the hard thresholding operator and $-\infty$ is set to a small enough number to ensure that the value after Softmax is 0. Adopting $\boldsymbol{S}_\delta^{(l)}$ instead of $\boldsymbol{S}^{(l)}$ in Eq. 5 enables us to filter out irrelevant tokens with small similarity values and only use highly similar image tokens to update the tokens in the token dictionary.

**ConvFFN.**  Recently, several works such as Zamir et al. (2022) start to add depth-wise convolutional layers and gating mechanism into feed-forward network. We simply adopt depth-wise convolution between two linear layers in FFN. It yields obvious improvement on ATD-light but little to ATD, since the extra receptive field brought by DWConv has a greater impact on shallow networks. The kernel size of DWConv are set as 7 for ATD-light and 5 for other versions of ATD.

### A.1.3 IMPLEMENTATION DETAILS

**ATD.**  We follow previous works (Liang et al., 2021; Chen et al., 2023) and choose DF2K(DIV2K (Timofte et al., 2017) + Flickr2K (Lim et al., 2017)) as the training dataset for ATD. We split the training process for ATD into two stages. In the first stage, we randomly crop $64 \times 64$ LR patches and the corresponding HR image patches for training. The batch size is set as 64, while commonly used Data augmentation tricks including random rotation and horizontally flipping are adopted in our training stage. We adopt the AdamW (Loshchilov & Hutter, 2018) optimizer with $\beta_1 = 0.9, \beta_2 = 0.9$ to minimize $L_1$ pixel loss between HR estimation and ground-truth. For the case of zooming factor $\times 2$, we train the model from scratch with an initial learning rate of $2 \times 10^{-4}$ for 600k iterations. We then finetune the $\times 4$ model based on $\times 2$ model for 300k iteration. The learning rate gradually decay to $1 \times 10^{-6}$ using cosine annealing scheduler (Loshchilov & Hutter, 2016). Then in the second stage, we utilize larger LR patches with size of $96 \times 96$ to further improve performance. The initial learning rate is reduced to $1 \times 10^{-5}$ for stable finetuning process of 50k iteration.

**ATD-light.**  To make fair comparisons with previous SOTA methods, we only employ DIV2K for training. Same as ATD, we train the $\times 2$ model from scratch and the $\times 4$ model is finetuned from $\times 2$ one. We increase the batch size to 128 for ATD-light and thus we can apply a large initial learning rate of $1 \times 10^{-3}$ for $\times 2$ training process. The training procedure for ATD-light is identical to that of ATD, except we don't apply the large-patch finetune stage for ATD-light.

**ATD-U.**  We choose ImageNet (Deng et al., 2009) as training data. To save training time, we first train ATD-U for 2000k iterations with a small window size of $8 \times 8$. Each batch consists of eight $128 \times 128$ noisy image patches. The initial learning rate is set as $2 \times 10^{-4}$ and we halve it at [400k, 800k, 1200k, 1600k, 1800k, 1900k]. Then we expand the window size to $16 \times 16$ and apply a two-phase finetuning strategy. The patch size is first enlarged to $256 \times 256$ for 160k iterations and further up to $512 \times 512$ for another 120k iterations. In the final finetuning phase we change the training dataset to DFWB(DIV2K, Flickr2K, BSD500 (Arbelaez et al., 2011) and WED (Ma et al., 2017)). The initial learning rate for each finetuning phase is decreased to $1 \times 10^{-5}$ for stability.

### A.2 ADDITIONAL EXPERIMENTAL RESULTS

### A.2.1 EXPERIMENTS ON GAUSSIAN COLOR IMAGE DENOISING

Additional experiments were carried out with ATD-U model for Gaussian color image denoising, and the experimental setting keeps consistent with grayscale one. We compare ATD-U with several SOTA methods including DnCNN (Zhang et al., 2017a), RNAN (Zhang et al., 2019), RDN (Zhang et al., 2018b), IPT (Chen et al., 2020), DRUNet (Zhang et al., 2021), SwinIR (Liang et al., 2021), Restormer (Zamir et al., 2022), and ART (Zhang et al., 2023). Quantitative results on Kodak24 (Franzen, 1999) and Urban100 (Huang et al., 2015) are provided in Table 5. Experimental results show that our proposed ATD-U yields 0.43dB and 0.18dB performance gain over Restormer and ART under severe noise level of $\sigma = 50$, which demonstrate the superiority of ATD-U to these methods.

Table 5: Quantitative PSNR(dB) comparison with state-of-the-art methods on **color image denoising** task. Best and second best results are colored with red and blue.

| Dataset | $\sigma$ | DnCNN | RNAN | RDN | IPT | DRUNet | SwinIR | Restormer | ART | **ATD-U** (ours) |
|---|---|---|---|---|---|---|---|---|---|---|
| **Kodak24** | 15 | 34.60 | - | - | - | 35.31 | 35.34 | 35.35 | 35.39 | 35.38 |
|  | 25 | 32.14 | - | - | - | 32.89 | 32.89 | 32.93 | 32.95 | 32.99 |
|  | 50 | 28.95 | 29.58 | 29.66 | 29.64 | 29.86 | 29.79 | 29.87 | 29.87 | 29.93 |
| **Urban100** | 15 | 32.98 | - | - | - | 34.81 | 35.13 | 35.13 | 35.29 | 35.36 |
|  | 25 | 30.81 | - | - | - | 32.60 | 32.90 | 32.96 | 33.14 | 33.25 |
|  | 50 | 27.59 | 29.08 | 29.38 | 29.71 | 29.61 | 29.82 | 30.02 | 30.27 | 30.45 |

### A.3 MORE VISUAL COMPARISONS

We provide more visual comparisons on image super-resolution in Fig. 7, Fig. 8, Fig. 9, Fig. 10 and grayscale image denoising in Fig. 11. These visual comparisons illustrate the potential of ATD and ATD-U in restoring sharp edge and texture under severe degradation.

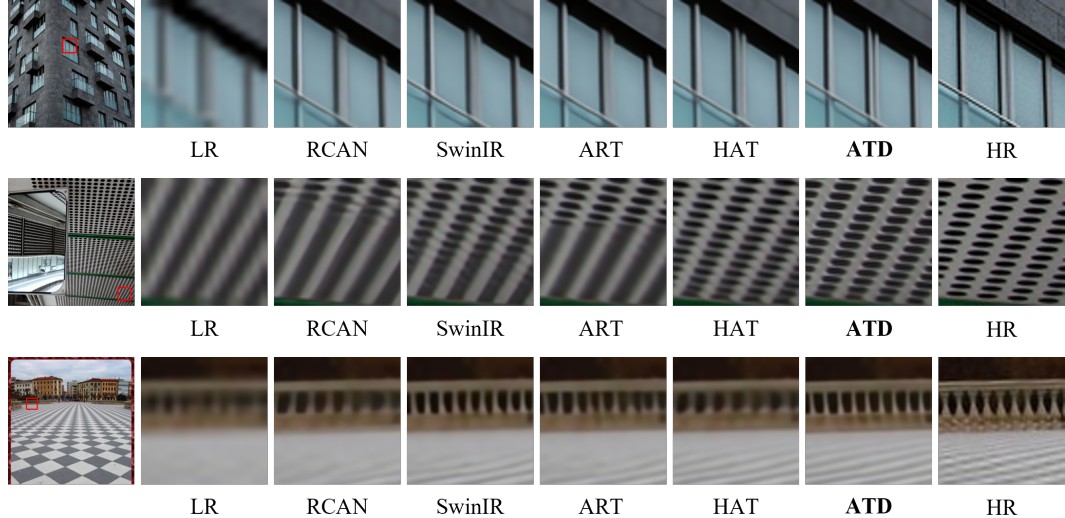

Figure 7: More visual comparisons of **classic image super-resolution** task on Urban100 (Huang et al., 2015) dataset. Test images from top to bottom are respectively "img_001", "img_004", "img_021".

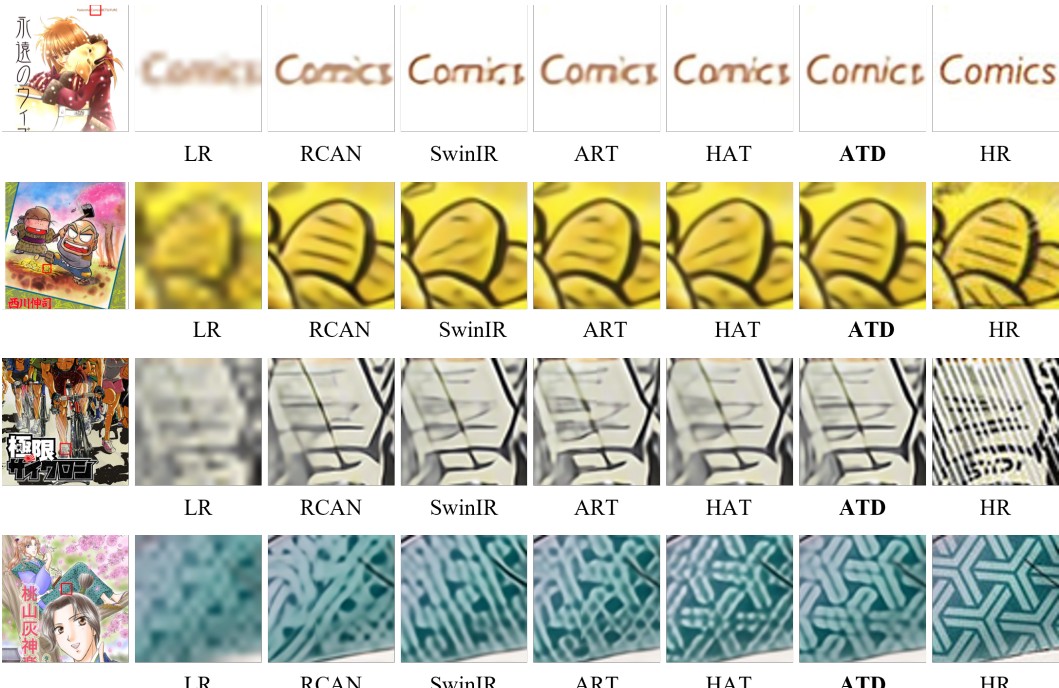

Figure 8: More visual comparisons of **classic image super-resolution** task on Manga109 (Matsui et al., 2016) dataset. Test images from top to bottom are respectively "EienNoWith", "JijiBabaFight", "KyokugenCyclone", "MomoyamaHaikagura".

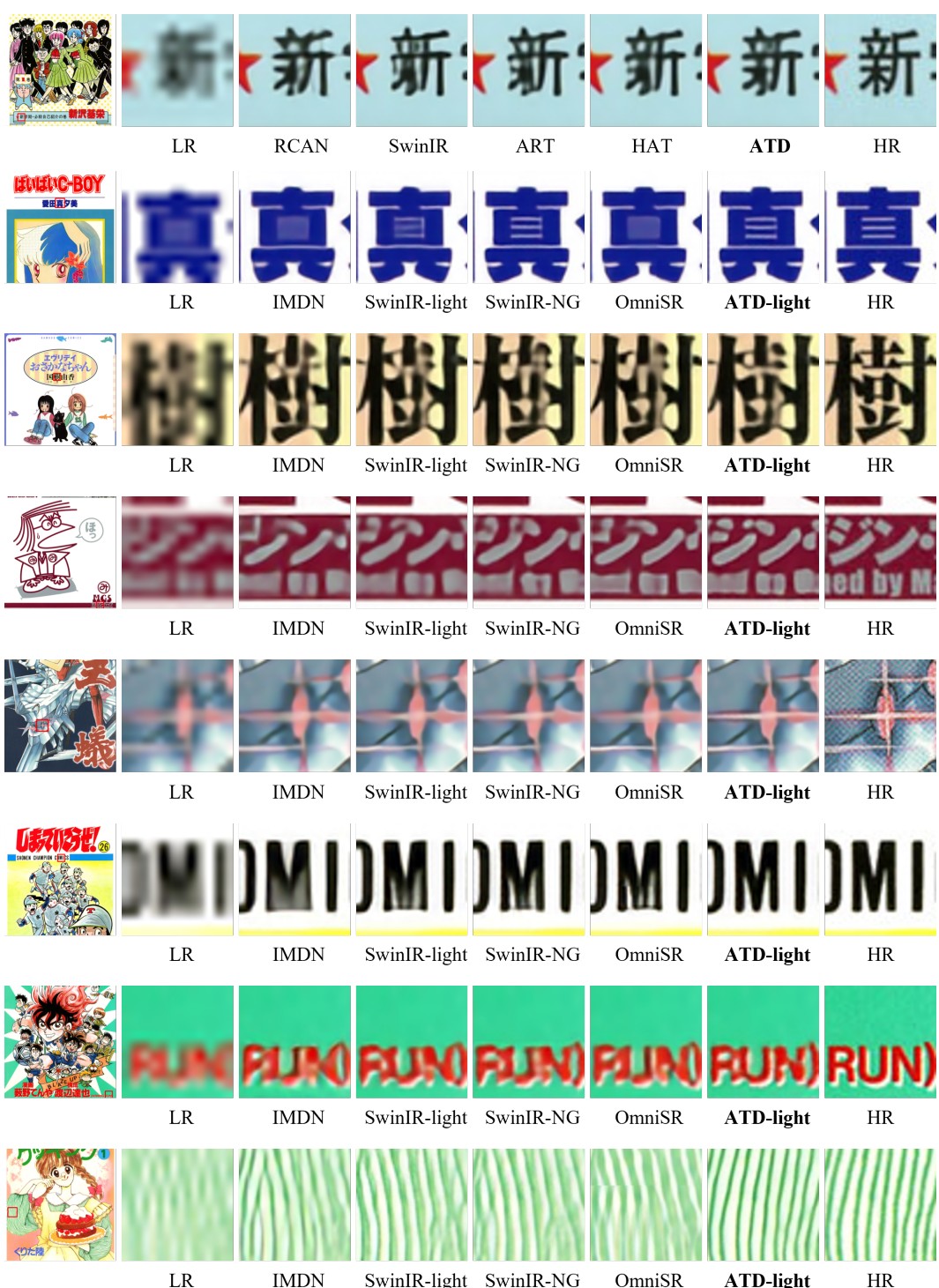

Figure 9: More visual comparisons of **classic image super-resolution** and **lightweight image super-resolution** task on Manga109 (Matsui et al., 2016) dataset. Test images from top to bottom are respectively "HighschoolKimengumi_vol01", "EverydayOsakanaChan", "Hamlet", "Joouari", "Shimattelkouze_vol26", "UltraEleven", "YumeiroCooking".

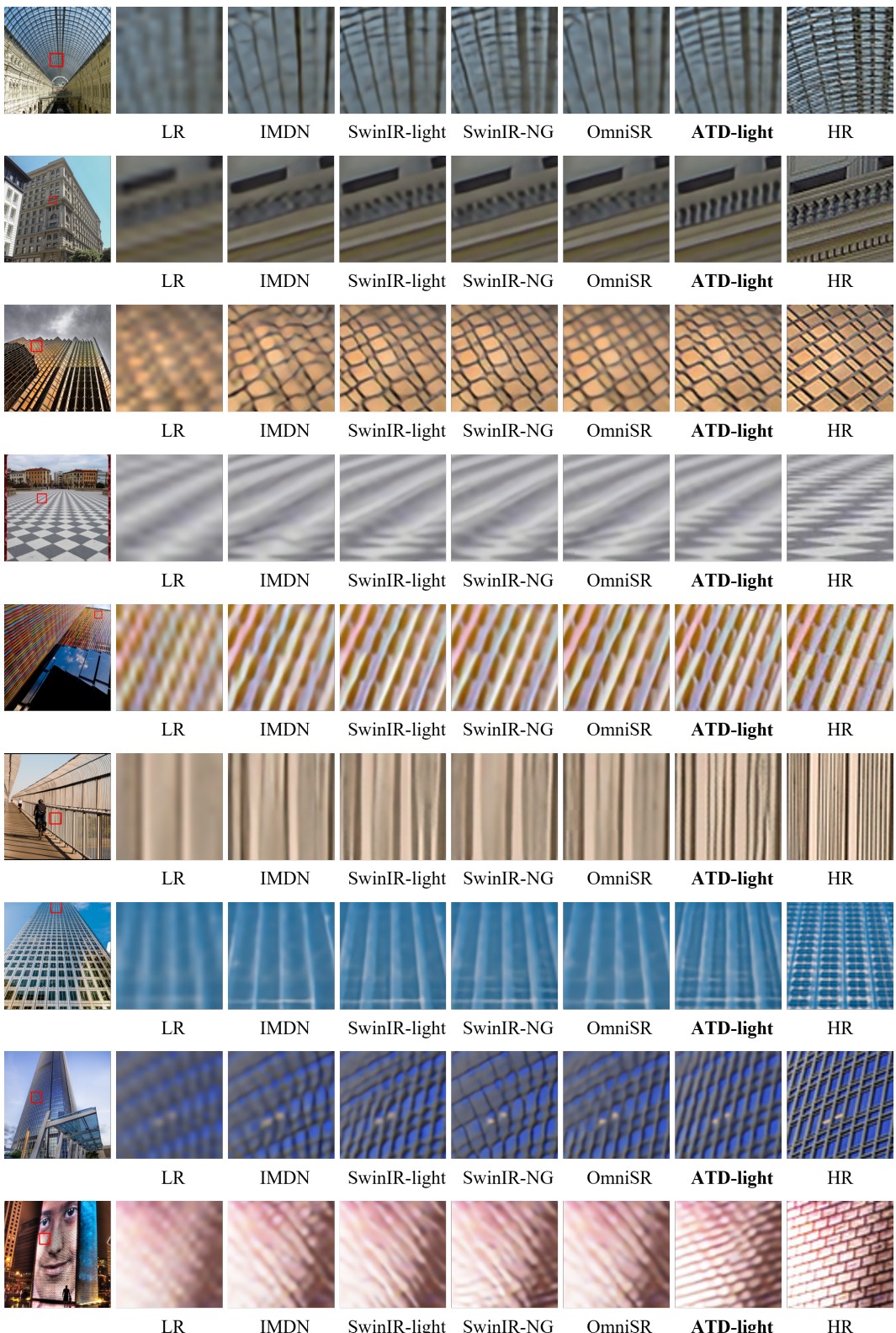

Figure 10: More visual comparisons of **lightweight image super-resolution** task on Urban100 (Huang et al., 2015) dataset. Test images from top to bottom are respectively "HighschoolKimengumi_vol01", "EverydayOsakanaChan", "Hamlet", "Joouari", "Shimattelkouze_vol26", "UltraEleven", "YumeiroCooking".

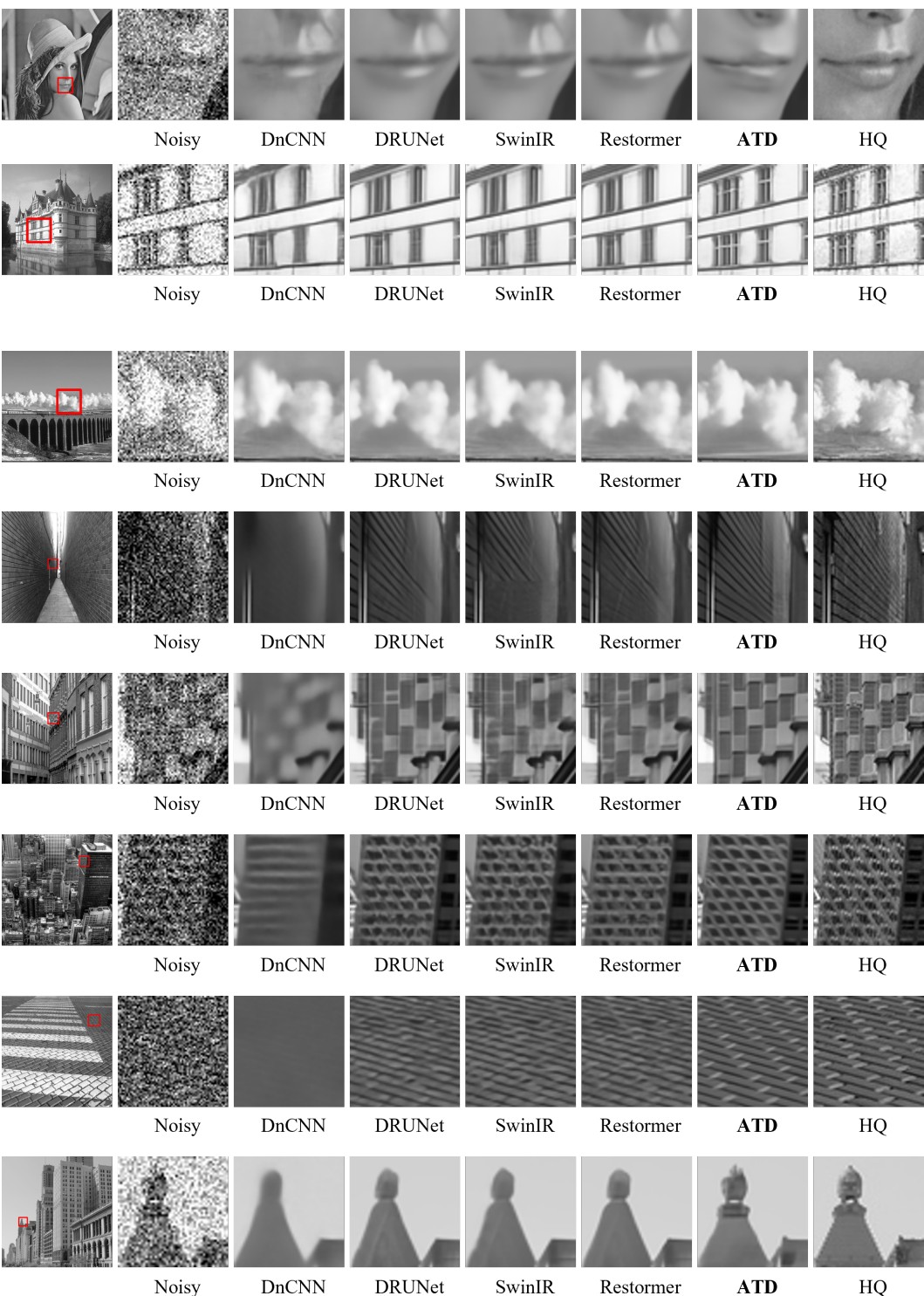

Figure 11: More visual comparisons of **grayscale image denoising** task. Test images from top to bottom are respectively "Lena" from Set12 (Zhang et al., 2017a), "test_003", "test_033" from BSD68 (Martin et al., 2001), and "img_038", "img_064", "img_073", "img_091", "img_097" from Urban100 (Huang et al., 2015) dataset.

