# OpenReview forum: "Enhancing Image Restoration Transformer with Adaptive Token Dictionary"
_ICLR.cc/2024/Conference — ICLR 2024 Conference Withdrawn Submission_

### Official Review · Reviewer_dEUU · 2023-10-29

**Soundness:** 2 fair
**Presentation:** 2 fair
**Contribution:** 1 poor
**Rating:** 3
**Confidence:** 5

**Summary:**

This paper proposes to enhance the Transformer model for image restoration through an Adaptive Token Dictionary (ATD). The methodology incorporates an external prior into its attention mechanism using a token dictionary. Furthermore, it introduces an adaptive dictionary refinement mechanism fuse the attention from shallow to deep layers. The paper provides experimental results across various benchmarks for image restoration tasks.

**Strengths:**

1. The writing is clear and easy to read.
1. Experiments on different benchmarks and tasks are provided to evaluate the method.

**Weaknesses:**

**1. The key concept "external prior" are not well defined and justified.**
According to the related works, the two prior concepts are taken from the paper of `(Wang et al., 2015)`, then the term "external prior" should refer to knowledge learned from the dataset, and "internal prior" should refer to self-similarity. If this is the case, every other IR networks that are trained from a dataset should have learned the "external prior". However, the paper classify some learning based approaches (SwinIR, HAT) as "internal prior", which is quite confusing. Moreover, **in terms of the "prior" classification**, the proposed method has no difference with these learning based methods (SwinIR, HAT, etc). Finally, there is no discussion and visualization about what is leaned in the "external prior".

**2. There is little novelty and technical contributions.**
There are many similar works that incorporate extra latent vectors into the Transformer model, such as Perceiver [a], visual prompt tuning [b], ViT-Adapter [c], and so on. The proposed external prior has little difference with these works.

**3. The second contribution "refining dictionary during testing phase" is not clearly explained.**
The "adaptive refinement" seems to be corresponding to the Sec 3.3. However, it looks like simple feature propagation from shallow to deep layers. How is the dictionary being refined to a specific image? Can it help to improve performance on specific images?

**4. The performance improvement is quite limited, especially when compared with HAT. Similar as other experiments on denoise and jpeg.**

**5. In ablation study, the result differences between different settings are not significant.** The baseline result is already good, and it is hard to see the improvement of the proposed "external dictionary".

**6. Text formats**: too many unnecessary upper cased words in the paper, for example "Dictionary", "Key", "Value".

**In conclusion, this paper has little technical contributions and the motivation is also unclear. The key contributions are based on subjective analysis and lacks experimental support. Therefore, I vote for reject.**

[a] Perceiver IO: A General Architecture for Structured Inputs & Outputs. Jaegle et al. ICLR2022.
[b] Visual Prompt Tuning. Jia et al. ECCV2022.
[c] Vision Transformer Adapter for Dense Predictions. Chen et al. ICLR2023.

**Questions:**

Please see weakness points above

---

### Official Review · Reviewer_mhzH · 2023-10-30

**Soundness:** 2 fair
**Presentation:** 2 fair
**Contribution:** 2 fair
**Rating:** 5
**Confidence:** 4

**Summary:**

This paper proposes an image restoration Transformer with a new adaptive token dictionary module. This adpative dictionary aims to encode the external priors, and enhance network features by using cross-attention. The method is inspired by dictionary learning. Experiments are conducted on several image restoration tasks including image super-resolution, denoising and JPEG compress artifact reduction.

**Strengths:**

1. The motivation of this paper is clear and sound.
2. Overall experimental results look good.

**Weaknesses:**

1. Important information are missed. How is $D_0$ generated? It is mentioned in Sec.3.3 that $D_0$ is introduced in Sec.3.2, while I cannot find an informative description there.
2. The proposed method is questionable. If the dictionary records an external prior, it should be a sparse representation obtained based on the entire dataset. However, the dictionary is optimized synchronously with the training process for each batch. It is weird.
3. The underlying logic of the effectiveness of external prior is not clear. For a powerful network, the meaning of external prior seems to be some image-specific high-frequency information. It is questionable how much effective information a limited dictionary representation can store. What is the length of $D$?

**Questions:**

See weakness.

---

### Official Review · Reviewer_Tpyp · 2023-10-31

**Soundness:** 2 fair
**Presentation:** 2 fair
**Contribution:** 2 fair
**Rating:** 5
**Confidence:** 2

**Summary:**

The paper proposes a new transformer-based architecture called Adaptive Token Dictionary (ATD) designed for restoring images in tasks such as super-resolution, noise reduction, and the reduction of JPEG artifacts. The key idea is to introduce an external learned "token dictionary" that provides supplementary information to the network, in addition to the standard self-attention that models internal image priors. Cross-attention is used between input features and the dictionary to select relevant dictionary tokens and reconstruct enhanced features. An adaptive dictionary refinement (ADR) method is proposed to customize the generic dictionary tokens to the specific test image using the similarity map from previous layers. This refines the dictionary during inference. Ablations show the token dictionary and ADR improve performance over baseline. ATD models achieve SOTA results on various benchmarks, outperforming recent transformer methods like SwinIR and HAT.

**Strengths:**

- Incorporating explicit external priors via a token dictionary in transformers seems reasonable for improving the restoration performance. It seems more interpretable than purely implicit priors.
- The ADR method to adapt the dictionary to image specifics is intuitive and elegant. Refinement using previous similarity maps seems interesting.
- Improves over baseline transformer and achieves new SOTA results on various image restoration tasks.
- Balances use of internal and external priors effectively.

**Weaknesses:**

- The writing of the method section seems not friendly to understand.
- The initial dictionary tokens are randomly initialized, which seems suboptimal. The learning details and the visualization analyses seem not enough to support the motivation.
- Ablations only show token dictionary and ADR improves over baseline. More ablation studies validating design choices like token size, reduction ratio, etc. would be useful.
- The related works are not enough. There are many dictionaries or codebook-based image restoration methods. However, this work did not discuss and compare with them. This makes the paper lack novelty, especially the claim that introduces a token dictionary learning method. This is not novel.
- The adaptive token dictionary refinement seems not reasonable enough. Because if the updated features x^{l+1} can provide sufficient features or details like a dictionary, the reconstruction of only using it can generate plausible results. So the usage of x^{l+1} for dictionary seems not reasonable.

**Questions:**

- How does the dictionary learn? Is it a learnable parameter or embedding? The existing methods usually adopt a pre-trained codebook using VQGAN for improving the restoration performance. Is there any difference between them in improving the performance? The learned codebook contains many structures prior, which can be used for all restoration tasks, while this work should be trained for different tasks.
- What does the learned dictionary represent? Does it mean the missing structure details or other features of the natural images? It is better to visualize it for better analysis.
- How sensitive is performance to token dictionary size? Is there a sweet spot balancing accuracy and efficiency?
- Does a pre-trained dictionary on a large dataset boost performance compared to random initialization?
- How do the learned dictionary tokens evolve during training? Do they converge to represent common patterns?
- Have you experimented with different token mixing strategies instead of simple summation after self-attention and dictionary attention?

---

### Official Review · Reviewer_wYuy · 2023-10-31

**Soundness:** 2 fair
**Presentation:** 2 fair
**Contribution:** 2 fair
**Rating:** 5
**Confidence:** 4

**Summary:**

This paper works on image restoration and the design of transformer-based networks. It proposes to use dictionary learning to capture external priors. The learnable tokens serve as the dictionary, which enhances the features through cross-attention and are updated via the proposed adaptive dictionary refinement. Experiments on image super-resolution, image denoising, and JPEG compression show the effectiveness of the proposed ATD network.

**Strengths:**

- The paper revisits the internal and external priors in image restoration.
- The paper proposes to use dictionary learning and cross-attention to enhance the features.
- An adaptive dictionary token update technique is proposed.
- Experiments show some advanced results for image super-resolution, image denoising, and JPEG compression artifact reduction.

**Weaknesses:**

- The method of using dictionary learning for image restoration is not novel. Some preview works also use similar techniques. There is a lack of thorough discussion in the context of dictionary learning with deep neural networks.
- It is not clear about the effectiveness of the proposed dictionary learning empirically and analytically. The ablation is only conducted for one specific setting. No computational tradeoffs are illustrated.
- Experiments show some comparable performance with existing methods.
- The paper should include more papers for discussion and more recent methods for comparisons, e.g., GRL and NAFNet.

Zheng, Hongyi, Hongwei Yong, and Lei Zhang. "Deep convolutional dictionary learning for image denoising." CVPR. 2021.\
Valanarasu, Jeya Maria Jose, Rajeev Yasarla, and Vishal M. Patel. "Transweather: Transformer-based restoration of images degraded by adverse weather conditions." CVPR. 2022.\
Zhang, Jinghao, et al. "Ingredient-Oriented Multi-Degradation Learning for Image Restoration." CVPR. 2023.\
Chen, Liangyu, et al. "Simple baselines for image restoration." ECCV. 2022.

**Questions:**

- The authors should investigate if the dictionary tokens represent different patterns, i.e., consistency of activation map for the dictionary tokens across images.
- There is no ablation for the dictionary choices, e.g., the number of tokens.
- Comparisons / visual examples for real data are expected.
- It is better to make the notations the same in Eq. (5) and Figure 1 (b).
- The format of Figure 5 should be refined.
- Best and second best results are colored with red and blue. (Table 4).
- Some typos in the text. Repeat “the” in “Specifically, the the attention map” (page 2). Two method motivates us (page 3). e believe (page 8).